# Precision Medicine: Technological Impact into Breast Cancer Diagnosis, Treatment and Decision Making

**DOI:** 10.3390/jpm11121348

**Published:** 2021-12-10

**Authors:** Tatiana Martins Tilli

**Affiliations:** Translational Oncology Platform, Center for Technological Development in Health, Oswaldo Cruz Foundation, Rio de Janeiro 21040-900, Brazil; tatiana.tilli@fiocruz.br; Tel.: +55-21-3882-9234; Fax: +55-21-2290-0494

**Keywords:** breast cancer, precision medicine, technology, diagnosis, treatment

## Abstract

Breast cancer is the most common cancer in women, impacting 2.1 million women each year. The number of publications on BC is much higher than for any other type of tumor, as well as the number of clinical trials. One of the consequences of all this information is reflected in the number of approved drugs. This review aims to discuss the impact of technological advances in the diagnosis, treatment and decision making of breast cancer and the prospects for the next 10 years. Currently, the literature has described personalized medicine, but what will the treatment be called for in the coming years?

## 1. Introduction

Breast cancer (BC) is the most common cancer in women and the second most common cancer overall, impacting 2.1 million women each year. In 2018, it was estimated that 627,000 women died from breast cancer. The percent surviving 5 year is 90.3% (2011–2017); however, almost all patients who develop metastatic disease will succumb to it. Figure 1 shows trends about different types of cancer, concerning numbers of publications and clinical trials. Among the tumors, the largest number of publications is about BC. The number of publications on BC is alarmingly greater than any other tumor type, this means that we researchers have studied and generated a lot of original and relevant information about these tumors. The question that remains is: What is the impact of all this information on clinical practice? The answer is evident when we compare the number of clinical trials (ongoing or complete) among the tumors. The number of clinical trials in breast cancer is absurdly higher than in other tumors. As a result, the impact of all this information is reversed directly to patients in the form of new FDA (Food and drug administration)-approved drugs (NCI). The number of drugs available for breast tumors is greater when compared to other tumors, with the exception of lymphomas. What is the impact of publications, clinical trials and FDA-approved drugs on patient survival? Only in the US, the death rate for breast cancer has decreased by 36% between 1989 and 2012. The decline in BC mortality has been attributed to both improvements in treatment and early detection in developing countries. However, it is not yet observable in low-middle income countries. Keeping track of the number of new cases, deaths, and survival over time (trends) can help scientists understand whether progress is being made and where additional research is needed to address challenges, such as improving screening or finding better treatments.

The main catalyst for the advance of knowledge in oncology was the technological progress, which has allowed us to understand in detail the biology of tumors at the single-cell level, but which has supported the advance in diagnosis, prognosis, therapy, and also decision making. In this review, we aim to discuss the relevance of technology in distinct aspects, such as detailed biological comprehension of BC, but also, molecular classification tools in the different areas of clinical practice, including personalized medicine, therapeutic decisions and follow-up practices based on tech advances.

## 2. Technologies and Databases

The first description of BC dating back 3000–2500 B.C. was made by Edwin Smith Surgical Papyrus. A BC case was counted as incurable if the disease was “cool to touch, bulging and spread all over the breast” [1,2]. Hippocrates’ theory in 400 B.C. of the unevenness of humours (blood, mucus secretion, and yellow and black bile) as a cause of cancer, and his classic descriptions of the progressive stages of breast cancer represent early hypotheses on the cause of cancer [3]. Galen, attributed BC’s cause to the accumulation of black bile in the blood, postulating that it was a systemic disease [4]. Additionally, ancient physicians postulated that the ending of menstruation was related to cancer due to the association of cancer with old age. In line with this theory, Galen allowed surgical wounds to bleed freely to get rid of the black bile and frowned on the use of ligatures. The word ‘crab’ for cancer was coined by him to illustrate the dilated veins radiating from the tumour [5].

One of the first and foremost treatments for BC is surgery. The turn of the century came to be synonymous with the name of William S. Halstead, Professor of Surgery at Johns Hopkins hospital in Baltimore, USA. His technique was called radical mastectomy (first reported around 1882–1894), with its emphasis on removing the majority of the breast tissues in one piece to prevent spread and removal of the pectoralis major to prevent recurrence, but often left patients with long-term pain and disabilities [6]. The hormone dependency of BC was initially hypothetical, through the observation that the disease was aggressive in younger women. Beatson started the era of endocrine surgery in 1906 [7]—before the discovery of estrogen receptors (ER) by Jensen in 1967 [8] and oopherectomy and adrenalectomy was introduced into clinical practice. These aggressive methods were gradually overtaken by ER modulators, luteinising hormone-releasing agonists and aromatase inhibitors.

Cytogenetic studies of the majority of all human solid tumors have lagged significantly behind the studies of human leucemias. Despite the relative paucity of cases studied from most human solid tumors, BC leads in the total cases studied due to high incidence. In summary, the cytogenetics studies showed hyperdiploidy, and deviation in chromosome numbers from normal diploid values apparently represented a poor prognostic variable [9,10]. Pandis and colleagues [11] described eight categories of karyotypes of BC: the structural rearrangements i(I)(q lo), der(I;I6)(qI0;plo), del(I)(q I I-I2), del(3)(p 12- I 3p 14-2I), and de1(6)(q2I -22) and the numerical aberrations +7, +18, and +20. These data confirm that BC is not a single disease, but is instead a collection of diseases that have distinct histopathological features, genetic and genomic variability, and diverse prognostic outcomes.

The Human Genome Project (HGP) was done by an international team of researchers looking to sequence and map all of the genes of *Homo sapiens* [12]. Beginning on 1 October 1990 and completed in April 2003, the HGP gave us the ability, for the first time, to read 95% of gene-containing parts of human sequence finished to 99.99% accuracy and made available to scientists and researchers. The data that have emerged since 2003 overwhelmingly support the value of this vision and have changed the way cancer is researched, understood and treated.

One of the first works published on this theme was that of Sjöblom and colleagues [13]. These authors analysed 13,023 genes in 11 breast and 11 colorectal cancers, which revealed that individual tumors accumulate an average of approximately 90 mutant genes but that only a subset of these contributes to the tumorigenesis and tumor progression process. Using stringent criteria to delineate this subset, 189 genes were identified as mutated at a significant frequency. A subsequent follow-up was published in 2007 [14] where the authors describe statistical and bioinformatic tools that could help the identification of mutations with a role in tumorigenesis. These results have implications for understanding the nature and heterogeneity of breast and colon human cancers and for using personal genomics for tumor diagnosis and therapy. The first whole cancer genome to be sequenced was from acute myeloid leukaemia and its matched normal counterpart was obtained from the same patient’s skin by Ley and colleagues [15]. They discovered ten genes with acquired mutations; FLT3 and NPM1 were previously described mutations that are thought to contribute to tumour progression, and eight were new mutations present in virtually all tumour cells, which are CDH24 and PCLKC, G-protein-coupled receptors (GPR123 and EBI2), a protein phosphatase (PTPRT), a potential guanine nucleotide exchange factor (KNDC1), a peptide/drug transporter (SLC15A1) and a glutamate receptor gene (GRINL1B). 

The first BC tumor was sequenced by Shah and colleagues [16]. These authors used Next Generation Sequence (NGS) to sequence the genomes (>43-fold coverage) and transcriptomes of an estrogen receptor (ER)-positive metastatic lobular BC at depth. The authors compare mutations present in the metastasis and measured the frequency of these somatic mutations in DNA from the primary tumour of the same patient, which arose 9 years earlier. Five of the 32 mutations (in ABCB11, HAUS3, SLC24A4, SNX4 and PALB2) were prevalent in the DNA of the primary tumour removed at diagnosis 9 years earlier, six (in KIF1C, USP28, MYH8, MORC1, KIAA1468 and RNASEH2A) were present at lower frequencies (1–13%), 19 were not detected in the primary tumour, and two were undetermined. The combined analysis of genome and transcriptome data revealed two new RNA-editing events that encode the amino acid sequence of SRP9 and COG3. These data showed that single nucleotide mutational heterogeneity can be a property of low or intermediate grade primary BC and that significant evolution can occur during disease progression. Taking all these publications together, these data paved the way for fertile avenues for basic and translational research in BC biology.

At this scenario of genomics and cancer emerged several databases in order to compilate several patient’s genomics data in an offer to open-access data. The Cancer Genome Atlas (TCGA-https://portal.gdc.cancer.gov/, accessed on 11 October 2021) database has revolutionized the field, as it is possible to access genomic data and patients’ clinical and pathological data, including therapeutic regime, tissue slide stained with hematoxylin and eosin and the number of mutations for each patient. TCGA was launched on February 2005 which contributes enormously for the number of publications regarding BC. TCGA data unveiled and characterized several lessons about breast cancer. For instance, (i) BC can be classified into four major molecular subtypes: Luminal A, Luminal B, human epidermal growth factor receptor 2 (HER2)-enriched and Basal-like or triple negative. Additionally, each subtype is associated with a unique panel of mutated genes. (ii) Basal-like subtype shares many genetic features with high-grade serous ovarian cancer, suggesting that the cancers have a common molecular origin and may share therapeutic opportunities. 

The compilation of TCGA data from 8,897 cases from BC patients identified 21,058 mutated genes, and the top 10 mutated genes are TP53 (4%), PIK3CA (3.8%), TTN (2.73%), MUC4 (2.21%), MUC16 (1.69%), CDH1 (1.67%), GATA3 (1.58%), MUC2 (1.28%), KMT2C (1.14%) and MAP3K1 (1.02%). Additionally, 156,432 somatic mutations were identified including the type, consequences, number of affected cases and impact on survival of each of them. Somatic mutations in only three genes (TP53, PIK3CA and GATA3) occurred at levels of more than 10% across all BC subtypes. In total, based on TCGA data, more than 1600 likely driver mutations in 93 BC genes were identified [17]. However, there were numerous subtype-associated and novel gene mutations, including the enrichment of specific mutations in GATA3, PIK3CA, and MAP3K1 with the Luminal A subtype [17]. This raises the question of whether we need to shift our clinical focus from subtypes to the genomic level. There are examples of the target of interest and the molecular subtype overlapping, such as HER2 amplification in the HER2-positive subtype, or FGFR amplification in Luminal B subtype. However, there are other cases where the target of interest and the molecular subtype do not overlap. This is the case for instances of PIK3CA mutations, which are present in 45% of Luminal A, 29% of Luminal B, 39% of HER2-enriched, and 9% of triple negative BC (TNBC) but most probably have different roles in these tumors.

In addition to TCGA, there are a number of web resources where it is possible to access -omics data from BC patients that have been helpful to the scientific community in order to revolutionise the field as presented in Table 1. More than that, there are a number of graphical summaries available online, such as The cBioPortal for Cancer Genomics, UALCAN and TNMplot which are open-access, open-source resources for interactive exploration of multidimensional cancer genomics data sets. The goal of these portals is to significantly lower the barriers between complex genomic data and cancer researchers by providing rapid, intuitive, and high-quality access to molecular profiles and clinical attributes from large-scale cancer genomics projects, and therefore to empower researchers to translate these rich data sets into biologic insights and clinical applications.

Which practical way is easy and fast for the research community to answer questions regarding, which genes, across all studies, are recurrently mutated? At what rates and are there particular hotspots or mutational patterns that correlate with either BC subtype or point to potential therapeutic options, like the use of PARP inhibitors in tumors lacking robust homologous recombination? However, the interpretation of NGS findings is complex since the technique cannot define the prevalence of particular mutations nor make any inference of their predictive values. Identification of appropriate biomarkers suitable for clinical use and their prognostic and predictive values is also an issue, along with controlling for sample heterogeneity and population-specific differences. Initial studies also failed to take into account differences in mutation frequency (between patients and tumour types), gene expression level and replication time, resulting in spurious findings. Consequently, at present, the technology is generally only validated for research. Our expectation is that these issues could be solved in the next few years, especially because this technology has been broadly used.

## 3. Emerging Concepts from -Omics Data

### Breast Tumor Knowledge and Heterogeneity

Several groundbreaking papers on BC sequencing have been published recently, demonstrating a major revolution in the field. It has long been known that BC is actually a group of diseases with different genotypes and phenotypes, prognoses and responses to treatment. Recognition of the importance of the ER, PgR and the HER2 in BC, and the large scale use of immunohistochemistry (IHC), enabled almost every cancer centre in the world to differentiate BC patients into three major groups: the hormone receptor (HR) positive group (which expresses ER and/or PgR), the HER2 positive group (which expresses HER2 by IHC or amplification detected by fluorescence in situ hybridization (FISH) and the triple negative (TN, or basal-like tumors) group (which is negative for ER, PgR, and HER2). Later, the ER-positive group was subdivided into two distinct prognostic groups, Luminal A and Luminal B, based on the percentage of Ki-67 (marker of proliferation) or the presence of PgR. The next step in the evolution of tools able to differentiate BC into molecular subtypes was the advent of gene expression arrays, which simultaneously measure thousands of genes to create a molecular landscape of the tumours. In 2000, Perou and colleagues [18] were the first to show that the phenotypic diversity of BC is accompanied by a corresponding diversity in gene expression patterns, defined as molecular portraits, that can be captured using cDNA arrays. These authors, also, unveil molecularly the basal-like subtype.

The Cancer Genome Atlas Network publication [19] evaluated a diverse set of breast tumours which were assayed using six different -omics technology platforms. Individual platform and integrated pathway analyses identified many subtype-specific mutations and copy number changes that identify therapeutically tractable genomic aberrations and other events driving tumour biology. This work significantly extends our knowledge base to produce a comprehensive catalogue of likely genomic drivers of the most common BC subtypes related to TP53 pathway, PIK3CA/PTEN pathway and RB1 pathway.

Curtis and colleagues [20] generated a robust, population-based molecular subgrouping of BC based on multiple genomic views. Copy number aberrations (CNAs) and SNPs influenced expression variation, with CNAs dominating the landscape in cis and trans. This work provides a framework for understanding how CNAs affect gene expression in BC and reveals novel subgroups that should be the target of future investigation, for example high-risk 11q13/14 cis-acting subgroup which presents a steep mortality trajectory. Additionally, these data unveil a novel subgroup which presents a favourable prognosis subgroup that is devoid of CNAs.

Pereira and colleagues [21] combined copy number, gene expression and mutation profiles to provide a richer understanding of the genomic landscape of BC. These authors sequenced 173 genes in 2,433 primary breast tumours that have CNA, gene expression and long-term clinical follow-up data. This work identifies 40 mutation-driver genes, and determines associations between mutations, driver CNA profiles, clinical-pathological parameters and survival. Associations between PIK3CA mutations and reduced survival are identified in three subgroups of ER-positive cancer (defined by amplification of 17q23, 11q13–14 or 8q24). High levels of intra-tumour heterogeneity are in general associated with a worse outcome, but highly aggressive tumours with 11q13–14 amplification have low levels of intra-tumour heterogeneity. These results emphasize the importance of genome-based stratification of BC, and have important implications for designing therapeutic strategies. 

Lehmann and colleagues [22] characterize TNBC subtypes and preclinical models to targeted therapies. The authors analyzed gene expression profiles and performed cluster analysis. Based on clustering, these authors identified six TNBC subtypes displaying unique gene expression and ontologies, including 2 basal-like (BL1 and BL2), an immunomodulatory (IM), a mesenchymal (M), a mesenchymal stem-like (MSL), and a luminal androgen receptor (LAR) subtype. Additionally, the authors identified TNBC cell line models representative of these subtypes. Driver signaling pathways were pharmacologically targeted in these cell line models as proof of concept that analysis of distinct gene expression signatures can inform therapy selection. The BL1 subtype had higher expression of cell cycle and DNA damage response genes, and representative cell lines preferentially responded to cisplatin. The BL2 subtype had a higher expresion of EGF, MET, WNT and metabolism pathways. The IM subtype was enriched in gene expression for CTLA4, IL7 and 12, NFKB and JAK/STAT pathways. The M and MSL subtypes were enriched in gene expression for epithelial-mesenchymal transition, and growth factor pathways and cell models responded to NVP-BEZ235 (a PI3K/mTOR inhibitor) and dasatinib (an abl/src inhibitor). The LAR subtype includes patients with decreased relapse-free survival and was characterized by androgen receptor signaling and overexpression of a number of metabolism pathways. LAR cell lines were uniquely sensitive to bicalutamide (an AR antagonist). These data may be useful in biomarker selection, drug discovery, and clinical trial rational design that will enable the alignment of TNBC patients to appropriate targeted therapies.

Altogether, the different strategies used to classify patients have led to a multidimensional understanding of the complex molecular make-up of BC, but the major question is whether these efforts have an impact in clinical practice, or how we can transpose all this information to the bedside. 

## 4. Era of Personalized Medicine

### 4.1. Targetable Genomic Alteration

In a sharp way to address this remaining question, a number of oncology hospitals structured a Precision Medicine Lab in order to sequence their patients, and also created a Tumor Boarding Staff, which is a genomic review board that included bioinformaticians, biologists, and clinicians to make decisions about patients by patient management based on targetable genomic alteration and clinical data. Progress in cancer genomics has raised hopes of increased precision in the identification of patients suitable for targeted therapies based on their genotypes.

Memorial Sloan Kettering Cancer Center (MSKCC) implemented a center-wide precision medicine strategy [23], which is a true multidisciplinary effort and comprehensive alignment of broad screening strategy with a clinical research enterprise that can use these data to accelerate the development of new treatments. The researchers developed and implemented MSK-IMPACT, a hybridization capture-based NGS panel capable of detecting all protein-coding mutations, CNAs, and selected promoter mutations and structural rearrangements in 410 cancer-associated genes in more than 10,000 metastatic cancer patients, of these 1237 from BC [24]. Through this panel, MSKCC efforts have produced an unparalleled dataset of matched tumor and normal DNA sequences from advanced cancer patients with associated pathological and clinical data. Based on this kind of strategy, it is possible to accelerate the development of genomically selected treatments which requires the availability of clinical studies to treat the large number of patients with potentially targetable alterations identified through the screening efforts. To accomplish this, MSKCC increased in 11% access to targeted therapies in clinical trials, because of their high efficacy in identifying molecularly enriched populations. In addition, individual disease specialty teams have opened studies for recurrent actionable genomic alterations observed in their respective patient populations. The presence of a universal screening program such as MSK-IMPACT onsite has made such studies increasingly feasible because they do not need to rely on slower and tissue-intensive single gene screening assays run at a central reference laboratory to identify rare genomic subpopulations within a particular histology.

Another example of an initiative designed on this basis is AURORA, pan-European molecular screening programme for metastatic BC (MBC) patients. This programme was developed by Breast International Group (BIG) in which approximately 1300 patients with MBC will consent to donate archived primary tumour tissue, as well as tissue collected from the biopsy of metastatic lesions and blood [25]. Tissues have been subjected to NGS performed on a panel of more than 400 cancer-related genes. Patients could be either treated per standard local practice or, if eligible, enrolled in one of the innovative, genotype-driven clinical trials to assess molecularly targeted agents for MBC. Additionally, also, all patients will be followed up for a further 10 years. In addition to the whole MBC population, analysis has focused on relevant categories of clinical interest: De novo and bone-only MBC, endocrine resistance, patients treated with targeted agents (mTOR, CDK4/6, HER2 inhibitors), chemo-resistant TNBC and BC with late relapse. Clinically relevant molecular categories were defined based on annotated aberrations: putative mechanisms of resistance alterations (ESR1, FGFR1, RB1), activating drivers (ERBB2, PIK3CA, AKT1), somatic and germline alterations in DNA damage repair genes (homologous recombination, mismatch repair). The authors reported on subtype switching from primary BC to MBC, on molecular signatures, on genes and pathways disrupted in several of these categories, and on the added value of circulating DNA (ctDNA) profiling. Efforts like this will improve the knowledge of the molecular evolution of BC and will help to identify biomarkers of response and/or resistance to both commonly applied BC treatments and innovative targeted agents.

SAFIR01 [26] study included 407 patients with biopsy samples from MBC. CGH array and Sanger sequencing were feasible in 283 (67%) and 297 (70%) patients, respectively. A targetable genomic alteration was identified in 195 (46%) patients, most frequently in PIK3CA (74 [25%] of 297 identified genomic alterations), CCND1 (53 [19%]), and FGFR1 (36 [13%]). In total, 117 (39%) of 297 patients with genomic tests available presented with rare genomic alterations (defined as occurring in less than 5% of the general population), including AKT1 mutations, and EGFR, MDM2, FGFR2, AKT2, IGF1R, and MET high-level amplifications. Therapy could be personalised in 55 (13%) of 407 patients. Of the 43 patients who were assessable and received targeted therapy, four (9%) received therapy directed against EGFR, AKT2, IGF1R, or FGF pathway, and had an objective response, and nine others (21%) had stable disease for more than 16 weeks. Serious (grade 3 or higher) adverse events related to biopsy were reported in four (1%) enrolled patients, including pneumothorax (grade 3, one patient), pain (grade 3, one patient), haematoma (grade 3, one patient), and haemorrhagic shock (grade 3, one patient). The authors conclude that personalisation of medicine for MBC is achievable, including for rare genomic alterations.

The UC San Diego Moores Cancer Center PREDICT Experience also describes a fruitful implementation of personalized medicine [27]. In this observational study, 347 patients with solid advanced cancers, including breast, brain, gastrointestinal and others, and NGS results were evaluated. Outcomes for patients who received a “matched” versus “unmatched” therapy following their NGS results were compared. Eighty-seven patients (25%) were treated with a “matched” therapy, 93 (26.8%) with an “unmatched” therapy. More patients in the matched group achieved stable disease (SD) ≥ 6 months/partial response (PR)/complete response (CR), 34.5% vs. 16.1%, (*p* ≤ 0.020 multivariable or propensity score methods). Matched patients had a longer median progression-free survival (PFS; 4.0 vs. 3.0 months, *p* = 0.039 in the Cox regression model). The main objective of all these trials was to investigate the feasibility of molecular screening to identify potential candidates for entry into phase 1 and 2 trials, in order to establish an in-depth molecular pre-screening protocol. In this scenario it is possible to save lives and quality of life, but also to save money for the system and ensure access to treatment for patients. All these studies have exposed that classification of driver events by levels of evidence of biological activity to prioritise multiarm drug trials is an attractive approach, and one that might help reduce the risk of mistaking noise for signal. Additionally, it is possible to clearly see that treatment based on precision medicine is moving from solely tissue origin as the primary criteria to molecular alteration, recently named basket trials. In this type of clinical trial it is possible to test how well a new drug or other substance works in patients who have different types of cancer that all have the same mutation or biomarker. Additionally, basket trials may allow new drugs to be tested and approved more quickly than traditional clinical trials and it is may also be useful for studying rare cancers and cancers with rare genetic changes. For examples see NCT03982173 and NCT02568267.

### 4.2. Examples of Molecular Tools Which Are Developed after -Omics Data: Available Tests, Technical Issues and Feasibility

The Breast Cancer Index (BCI) test analyzes the expression of seven genes from formalin-fixed, paraffin-embedded breast tumor tissue by quantitative RT-PCR assay. Five of them are related to cell cycle (BUB1B, CENPA, NEK2, RACGAP1 and RRM2) and also two genes associated with tumor responsiveness to endocrine therapy, which are HOXB13 and IL17BR. This test helps predict the risk of node-negative or positive (with 1–3 positive nodes), HR+ early stage. The BCI reports two scores: how likely the cancer is to recur 5 to 10 years after diagnosis and how likely a woman is to benefit from hormonal therapy for a total of 10 years. Outcomes predicted:Risk of distant recurrence 5–10 years post-diagnosis and risk of benefit from 10 years adjuvant endocrine treatment. The BCI test is not approved by the FDA but may be covered by some insurance companies. Breast cancer index (HOXB13/IL17BR ratio (H/I)) was evaluated for its ability to predict the benefit from extended endocrine therapy in patients previously randomized in the Adjuvant Tamoxifen-To Offer More? (aTTom) trial [28]. BCI with high H/I expression was predictive of endocrine response and identified a subset of HR+, N+ patients with significant benefit from 10 versus 5 years of tamoxifen therapy. These data established level 1B evidence for BCI as a predictive biomarker of benefit from extended endocrine therapy.

The EndoPredict test integrates tumor biology and pathology to accurately predict early and late recurrence. The test is appropriate for pre- and post-menopausal women, HR+, HER2- early stage with node-negative or up to three positive lymph nodes. This assay can be performed on formalin-fixed paraffin-embedded tissue [29]. The EndoPredict test analyzes by quantitative reverse transcription polymerase chain reaction (qRT-PCR) of 12 genes, which are 3 proliferation-related genes (UBE2C, BIRC5 and DHCR7), 5 hormone receptor-related genes (STC2, AZGP1, IL6ST, RBBP8 and MGP), 3 normalization genes (CALM2, OAZ1 and RPL37A) and 1 for DNA contamination (HBB). The 12-gene molecular Score significantly improved prognostic performance when added to nodal status, tumor size, age and grade, in order to calculate an EPclin Risk Score that categorizes the cancer as having either a high or low risk of distant recurrence. Outcomes predicted: Risk of distant recurrence 10 years post-diagnosis (low or high). The EndoPredict test is sold as a kit to local pathology labs, rather than done as a centralized laboratory service like some of the other genomic tests. The EndoPredict test is not approved by the FDA. Filipits and colleagues [30] evaluated the prognostic value of a 12-gene expression assay (EndoPredict) in the combined ABCSG-6/8 cohorts with longer clinical follow-up. This study demonstrates that EndoPredict test can identify patients at low risk for early or late recurrence who may safely forgo adjuvant chemotherapy or extended endocrine therapy, respectively, regardless of nodal status.

The Oncotype DX test is used to estimate a woman’s risk of recurrence of early stage HR+ breast cancer, as well as how likely she is to benefit from chemotherapy after breast cancer surgery. The Oncotype DX breast cancer assays can help physicians and patients decide on the best course of treatment. The test is a reverse-transcriptase–polymerase chain-reaction 21-gene assay performed on RNA extracted from formalin-fixed paraffin-embedded tissue. The panel includes genes associated with proliferation (Ki-67, STK15, Survivin, Cyclin B1 and MYBL2), invasion (Stromelysin 3 and Cathepsin L2), Oestrogen (ER, PR, Bcl2 and SCUBE2), the human epidermal growth factor HER2neu (GRB7 and HER2) and GSTM1, BAG1 and CD68 genes. Relative expression of these genes is measured in relation to the average expression of five reference genes (Beta-actin, GAPDH, RPLPO, GUS and TFRC). An algorithm was designed to calculate a numerical Recurrence Score (RS) ranging between 1 and 100. A low RS value (RS < 18) corresponds to a low probability of distant recurrence at 10 years, whereas a higher score (RS ≥ 31) is associated with a higher probability. For invasive breast cancer, the Oncotype DX Breast Recurrence Score predicts chemotherapy benefit and the likelihood of distant BC recurrence [31]. Outcomes predicted: Recurrence Score (RS): Risk of distant recurrence 10 years post-diagnosis; Low:<18, intermediate: 18–30, high: >30. Oncotype DX is not approved by FDA. The Trial Assigning IndividuaLized Options for Treatment (Rx), or TAILORx, is a very large prospective phase III, academic, US-based trial designed to refine adjuvant chemotherapy withdrawal using the results of Oncotype DX. The primary question of the trial pertains to women harbouring axillary node-negative, early stage, HER2-negative, HR-positive BC who have an intermediate risk based on Oncotype DX score (RS 11–25), in whom the benefit of the addition of adjuvant chemotherapy to endocrine therapy is being investigated; the results for this cohort were published by Sparano and colleagues [32,33]. The authors showed (1) excellent outcome of women with a low-risk Oncotype DX score (RS < 11) and showed that following adjuvant endocrine therapy, 99% of these women remain free from recurrence at any site after five years of hormone therapy alone [32], and (2) adjuvant endocrine therapy and chemoendocrine therapy had similar efficacy in women with HR-positive, HER2-negative, axillary node-negative BC who had an intermediate score result, although some benefit of chemotherapy was found in some women 50 years of age or younger [33].

MammaPrint assay was developed as a prognostic tool, based on the 70-gene prognostic signature developed by van’t Veer and colleagues [34], to identify those BC patients that may safely undergo chemotherapy and to predict risk of BC metastasis. The 70 genes that make up the MammaPrint signature were selected from genome-wide expression data using a data-driven approach. This resulted in a set of 70 genes that was able to predict the risk of recurrence with high sensitivity. The genes included in the MammaPrint signature comprehensively measure the six hallmarks of cancer-related biology [35]. The test is a microarray-based gene expression profile, which showed two categories of tumors with different risks of developing a short interval distant metastasis in patients without tumor cells in local lymphnodes at diagnosis: low-risk tumors and high-risk tumors. Women in the low-risk group have a distant metastasis-free survival rate of over 90% without receiving chemotherapy [36]. Unlike the Oncotype DX assay, this test requires freshly prepared tissue collected in an RNA preservative solution. Additionally, MammaPrint is approved by the FDA. Outcomes predicted:Risk of distant recurrence 10 years post-diagnosis (low: <0.4, high: ≥0.4. Microarray In Node negative Disease may Avoid ChemoTherapy (MINDACT-EORTC Protocol 10041–BIG 3-04) is a prospective randomised study comparing Mammaprint with the common clinic-pathological criteria in selecting patients for adjuvant chemotherapy in BC patients with 0–3 positive nodes [37].MINDACT has enrolled 6600 patients and results showed excellent 5-year distant metastasis-free survival of 94.7% in patients with BC of high clinical and low genomic risk who did not receive chemotherapy. Recently, Piccart and colleagues [38] presented long-term follow-up results (~9 years) together with an exploratory analysis by age. Based on these data, Mammaprint assay shows an intact ability to identify among women with high clinical risk, a subgroup, namely patients with a low genomic risk, with an excellent distant metastasis-free survival when treated with endocrine therapy alone. Additionally, this benefit appears to be age-dependent, as it is only seen in women younger than 50 years.

A 50-gene qPCR assay (PAM50) was developed to identify gene expression-based subtyping and prediction of chemotherapy benefits using RNA isolated from FFPE tissue [39]. The PAM50 test was used to develop a prognostic risk of relapse score based on a proliferation score, which includes the quantification of the expression profile of genes related to cell cycle progression; and composite scores that include tumor size with molecular phenotypes. It is not approved by FDA. PAM 50 risk of recurrence score was accessed in 1478 postmenopausal patients of the ABCSG-8 trial treated with adjuvant endocrine therapy. The results of the primary analysis constitute Level 1 evidence for clinical validity of the PAM50 test for predicting the risk of distant recurrence in postmenopausal women with ER+. A 10-year metastasis risk of <3.5% in the ROR low category makes it unlikely that additional chemotherapy would improve this outcome. This finding could help to avoid unwarranted overtreatment [40]. Outcomes predicted: Intrinsic subtyping Risk of Recurrence (ROR) score (low, intermediate, high).

BluePrint is an 80-gene profile that classifies BC into molecular subtypes. The profile separates tumors into Luminal A and B, HER2 and triple negative subgroups by measuring downstream genes for each of these molecular pathways. This assay is a genomic signature by microarray-based RNA gene expression to guide the choice of therapies and combinations of therapies. This test is not yet approved by the FDA. By combining MammaPrint and BluePrint, patients can be classified into the following four subgroups: Luminal-type/MammaPrint Low Risk (Luminal A-type); Luminal-type/MammaPrint High Risk (Luminal B-type); HER2-type and Basal-type patients. BluePrint with MammaPrint molecular subtyping together helps to improve prognostic estimation and the choice of therapy versus immunohistochemistry/fluorescence in situ hybridization (IHC/FISH) [41,42].

BreastOncPx is a 14-gene signature assay (BUB1, CCNB1, CENPA, DC13, DIAPH3, MELK, MYBL2, ORC6L, PKMYT1, PRR11, RACGAP1, RFC4, TK1, UBE2S) that provides prognostics information for lymph node-negative, ER+ BC patients and is associated with risk of distant metastasis. It helps identify higher-risk patients who might benefit from additional therapy [43]. Outcomes predicted:Metastasis Score: Risk of distant recurrence 10 years post-diagnosis (low, moderate, high). It is not approved by the FDA.

Symphony provides complete tumor profiling and is used to support therapeutic choices for BC. Symphony is a compendium of four assays to support BC treatment decisions: (a) MammaPrint which determines the risk of recurrence; (b) BluePrint which determines molecular subtypes; (c) TargetPrint determines ER, PR and HER2 status; and (d) TheraPrint which identifies alternative types of therapy for metastatic disease. It is a microarray-based RNA gene expression methodology which uses formalin-fixed, paraffin-embedded tissue sections from BC patients and it is not approved by FDA.

More recently, FoundationOne CDx (Cambridge, MA, USA, 2021) was launched in 2010 for five tumor indications: breast cancer, ovary cancer, non-small cell lung cancer (NSCLC), melanoma and colorectal tumor. It is an NGS based in vitro diagnostic device for detection of substitutions, insertion and deletion alterations (indels), and copy number alterations (CNAs) in 324 genes (guideline-recommended genes) and select gene rearrangements, as well as genomic signatures including microsatellite instability (MSI) and tumor mutational burden (TMB). FoundationOne CDx is performed exclusively as a laboratory service using DNA extracted from FFPE tumor samples. Outcomes predicted: The test is intended as a companion diagnostic to identify patients who may benefit from treatment with the targeted therapies in accordance with the FDA-approved and therapeutic product labeling, and also, detects drug resistance mechanisms.

## 5. Clinically Translatable Promises: Next 10 Years

Precision oncology is an approach to cancer treatment that seeks to identify effective therapeutic strategies for every patient. The identification of mutations that arise during treatment that confer drug sensitivity is paramount for precision cancer care of patients with advanced disease. However, there remain a significant number of cases where genomic analysis currently fails to identify effective drugs or applicable clinical trials. Even when targetable genomic alterations are discovered, patients do not always respond to therapy. Strategies to confirm therapeutic efficacy or identify additional options would be beneficial to both clinicians and patients. To address this need, a number of researchers have established tumor organoids and AVATARs [44], which facilitates the integration of genomic data with drug screening of patients’ tumor samples in an iterative platform to identify effective therapeutic regimens for individual patients. Although it may not be feasible to utilize this approach for all patients with cancer, the integration of genomics with drug-sensitivity data across many tumor types may significantly affect patient outcomes in the future. Organoid technology is used in research as an intermediate model between cancer cell lines in vitro and xenografts as shown for colorectal, pancreatic, BC, and prostate cancers. This technique differs from traditional cell culture by maintaining cancer cells in three-dimensional (3-D) cultures, and then retaining cell–cell and cell–matrix interactions that more closely resemble those of the original tumor compared with cells grown in two dimensions on plastic. Utilizing established 3-D patient organoid culture systems, personalized high-throughput drug screening coupled with genomic analysis from patient-derived tumor samples offers a unique opportunity to stratify and identify effective cancer therapies for individual patients [45]. By adding a drug screening component into the precision medicine platform, this allows to (i) compare the response of individual tumors to specific drugs in order to provide individualized recommendations to help guide patient care; (ii) assess how individual tumors adapt in response to therapies and better understand the context in which these agents are efficacious; (iii) determine the next course of action for cases where standard clinical options have already been exhausted; (iv) create a database that relates drug sensitivity to tumor genetics to nominate potential therapeutic strategies even when only genomic data are available.

Sachs and colleagues [46] described a robust protocol for long-term culturing of human mammary epithelial organoids. Cancer organoides have been helped to explore the functional tests, genotype-phenotype correlations and drug discovery and selection of it [47,48] The authors generated more than 100 primary and metastatic BC organoid lines which broadly recapitulate the diversity of the disease. BC organoid morphologies typically matched the histopathology, hormone receptor status, and HER2 status of the original tumor. DNA copy number variations as well as sequence changes were consistent within tumor-organoid pairs and largely retained even after extended passaging. BC organoids furthermore populated all major gene-expression-based classification groups and allowed in vitro drug screens that were consistent with in vivo xeno-transplantations and patient response. This study describes a representative collection of well-characterized BC organoids available for cancer research and drug development, as well as a strategy to assess in vitro drug response in a personalized fashion.

The generation of tumor xenograft in zebrafish embryos and larvae, or zebrafish avatar, represents an opportunity to study tumor growth, angiogenesis, cell invasion and metastatic dissemination, the interaction between tumor and host in vivo avoiding immunogenic rejection, representing a promising platform for the translational research and personalized therapies [49]. This approach could be applied using both breast cancer cell lines and a patient’s biopsy to perform drug testing.

The main efficacy limitations of drugs currently used to treat cancer include insolubility, systemic toxicity, and drug resistance, compounded by debilitating side effects such as nausea, fatigue, neuropathy, and organ failure. An effective solution to overcome these clinical limitations is nanomedicine. Nanomedicine enables a series of advances in order to improve the properties of drugs, such as solubility and/or stability, and also offers new therapeutic approaches through the development of drugs capable of providing ways of direct and deliver drugs to the body with more precision, safety and efficacy, as well as the development of promising strategies for diagnosis, prognosis and cell and tissue regeneration [50,51,52]. Advances in nanotechnology have led to the development of new nanomaterials whose physicochemical properties differ from their larger counterparts due to their higher surface/volume ratio, making them excellent candidates for biomedical applications [53]. Together, the different nanomedicine approaches directly contribute to knowledge about tumor biology and heterogeneity, both inter- and intra-tumor for potential personalized treatment [54,55]. Therapeutic siRNAs have been considered one of the most significant discoveries for pharmaceutical development considering the marketing aspect as well [56]. This approach has been widely used for the validation of therapeutic targets and consequently generates intense activity in the industrial development of these inhibitors [57,58,59]. According to the analysis of US patents, the most studied disease using the siRNA strategy is cancer. Recently, many types of siRNA have been evaluated in clinical trials with safe and effective results [60,61]. However, FDA approval for marketing the first siRNA-based drug took place only last three years [62]. The drug called Onpattro (Patisiran), from the Alnylam company, is a milestone in the development of these therapeutic products and reinforces that therapies with siRNA are a scientific and market trend.

However, due to the low stability of siRNA molecules, injecting nucleic acids directly into the bloodstream is not recommended [63]. These can be degraded by endonucleases leading to rapid elimination from the organism. Furthermore, most siRNAs are anionic and therefore do not cross the cell membrane which is also negatively charged. Thus, nanotechnology emerges as a delivery strategy for siRNA, so that it can remain stable for longer in the body [50]. Ideally, a nanocarrier should be stable in circulation, to protect and deliver its therapeutic load (drug) to the target tissue, have good penetration and retention in the target tissue so that drug release occurs within the therapeutic window, and ultimately, be organically excreted to prevent long-term accumulation toxicity. In the case of cancer, drugs could be targeted to tumors, avoiding systemic side effects of traditional therapies [50]. Although nanocarrier technology has advanced, the lack of target specificity limits systemic use. The functionalization of the surface of the nanoparticles through the fixation of a ligand that interacts with specific receptors of a given tissue can optimize the administration of the active, transporting it to the desired site of action in a selective way [64]. Davis and colleagues [65] present the first results of the functional siRNA system in humans, through systemic administration of second-generation nanoparticles and treatment of solid tumors. The nanocarrier system could also be implemented for a substantial number of chemotherapeutic drugs with well-understood molecular mechanisms of action and FDA-approved.

In tumor cells, the levels of some surface markers may increase, undergo modifications, or new markers may appear on the cell surface. All of this can be used to differentiate healthy from tumor cells [66,67]. Some proteins are already well known in cancer studies in relation to molecular diagnosis and treatment with monoclonal antibodies, for example, estrogen and progesterone receptors, vascular endothelial growth factor receptor (VEGFR), EGFR and HER2 [66]. Approximately 70% of drugs approved by the FDA and in the clinical phase are directed to plasma membrane proteins, as they are more accessible targets than intracellular proteins. Furthermore, many oncogenic processes take place primarily in the plasma membrane, including proliferation, adhesion and migration [68]. In this context, these proteins have distinct clinical relevance: (A) Differentially expressed membrane proteins could be used as specific tumor biomarkers for tumors, this would allow the molecular diagnosis of the disease. (B) Membrane proteins could help to define a neoplasia-free surgical margin, which means the boundary between tumor tissue and healthy tissue, which is an important clinical limitation. The characterization of the disease-free surgical margin has an influence on potential recurrences, metastasis formation and unnecessary esthetic changes. (C) These membrane proteins, when anchored in nanoparticles, present the objective of targeted delivery of therapeutic load, considering the importance that this treatment would be directed only to tumor cells and would not reach healthy cells, differently from what happens in the chemotherapy treatment used in clinical practice. (D) Another treatment approach would be monoclonal antibodies or aptamers that would bind to a specific target, in this case, identified proteins that are overexpressed in tumors, preventing the activation or recognition of these receptors by other molecules, consequently inhibiting tumor processes.

There are 48 studies at starting point or ongoing registered at clinicaltrials.gov related to liquid biopsies and BC, which are regarding new circulating biomarkers for triple negative tumors, detection of Circulating Tumor Cells (CTCs) and Cell Free DNA (cfDNA) in peripheral blood of BC patients to develop the clinical application for early detection and diagnostics (NCT03511859), serial circulating tumor DNA (ctDNA) monitoring during adjuvant capecitabine in early triple-negative BC (NCT04768426), characterization and comparison of druggable mutations in primary and metastatic tumors, CTCs and cfDNA in MBC patients (MIRROR) (NCT02626039), studies of metabolites markers, response to treatment for ER+ patients. The next 5 years will be promising in terms of applying this knowledge about CTCs and cfDNA in the clinic.

Other technical advances that occurred in BC treatment are related to therapeutic vacines and immunotherapies. BC is considered a cold tumor, which means that it describes a tumor that is not likely to trigger a strong immune response. Cold tumors tend to be surrounded by cells that are able to suppress the immune response and keep T cells (a type of immune cell) from attacking the tumor cells and killing them (NCI definition). For this reason, they are a challenge for therapy and the development of a therapeutic vaccine. However, there are 223 clinical trials completed or ongoing registered at clinicaltrials.gov. As an example, vaccination using E75 which is an immunogenic peptide from the HER2/neu protein that is highly expressed in BC, presents promising results (NCT00854789). Peoples and colleagues [69] conducted a clinical trial of an E75 + granulocyte-macrophage colony-stimulating factor vaccine to assess safety, immunologic response, and the prevention of clinical recurrences in patients with disease-free, node-positive breast cancer. Fifty-three patients with NPBC were enrolled and HLA typed. HLA-A2+ patients (*n* = 24) were vaccinated, and HLA-A2- patients (*n* = 29) are observed prospectively as clinical controls. Only minor toxicities have occurred (one grade 3 [4%]). All patients have demonstrated clonal expansion of E75-specific CD8+T cells that lysed HER2/neu-expressing tumor cells. Patients have developed delayed-type hypersensitivity reactions to E75 post vaccination compared with controls (33 vs. 7 mm). The disease-free survival in the vaccinated group is 85.7% compared to 59.8% in the controls at 22 months’ median follow-up with a recurrence rate of 8% compared to 21%, respectively. Median time to recurrence in the vaccinated patients was prolonged (11 vs. 8 months), and recurrence correlated with a weak delayed-type hypersensitivity response. In summary, HER2/neu (E75) vaccine was safe and effective in eliciting a peptide-specific immune response in vivo. Induced HER2/neu immunity seems to reduce the recurrence rate in patients with node-positive breast cancer.

Qi and colleagues [70] assess the immune response, disease progression, and post-treatment survival of ER/PR double-negative stage II/IIIA BC patients vaccinated with autologous dendritic cells (DC) pulsed with autologous tumor lysates. DC vaccines elicited Th1 cytokine secretion and increased NK cells, CD8+ IFN-+ cells but decreased the percentage of CD3+ T cells and CD3+ HLA-DR+ T cells in the peripheral blood. There was no difference in overall survival between the patients with and without DC vaccine. The 3-year progression-free survival was significantly prolonged: 76.9% versus 31.0%.

The development of imunotherapies based on immune-check inhibitors has resulted in a paradigm shift for immune oncology therapeutics in the past few decades. Blocking antibodies against programmed death 1 (PD-1) and programmed death ligand 1 (PD-L1) leads to tumor regression and a long-lasting response in patients with various tumors that are non-responsive to standard treatments [71]. BC is considered poorly immunogenic and present lower mutation burden compared to ‘inflamed’ tumors, as melanoma and non-small cell lung carcinoma. Accumulated results have demonstrated that higher T lymphocyte infiltration was observed in TNBC compared to other molecular subtypes; its use was approved in combination with chemotherapy for first-line therapy in metastatic TNBC overexpressing PD-L1.

As immune checkpoint inhibitors alone have modest clinical activity in advanced BC, there is a growing interest in combinatorial modalities, and particularly for their rapid development in the early disease setting. There are 324 studies registered at clinicaltrials.gov regarding combinatorial immunotherapies, and the rationale behind that is to increase the clinical benefit of immunotherapy against BC. In the era of precision medicine, exploration of actual immune response and development of biomarkers is required to maximize the clinical benefit of cancer immunotherapy.

Technology has already improved BC diagnosis, treatment and is helpful in decision making. In recent years, technological advancement was focused on identifying and characterizing mutations in tumors; however, current technological approaches and recent patents filed point to an emergent scenario of molecular panels that include messenger RNA expression fluctuations, patterns of epigenetic alterations, pharmacogenomics data in order to identify responders and non-responders to therapeutic regimens, drug repositioning and approaches related to the identification of biomarkers or CTCs in biological fluids with the perspective of being less invasive and aggressive to the patient. The next 10 years should be promising and profitable for BC and MBC treatment.

## 6. Conclusions

In the last 10 years, technology has greatly impacted diagnosis, therapy and helped therapeutic decision-making in the area of breast cancer. Ease of access to clinical and genomic data supported and fostered the great therapeutic and prognostic advance. Personalized medicine has evolved and has solid foundations to continue this growth on a logarithmic scale. In this review, key strengths were highlighted, but new issues emerge in this scenario. What are the roadblocks to having all cancer care settings use multidisciplinary tumor boards to guide patient care? What needs to be in place over the next 10 years for this to become a reality? Especially in low-middle income countries, the answer for these questions is far from reach. It is urgent to discuss the accessibility of new treatments and molecular panels. How do we implement expensive treatments such as immunotherapy in clinical practice? For example, MSKCC increased in 11% access to targeted therapies in clinical trials, because of their high efficacy in identifying molecularly enriched populations. The presence of a universal screening program will be helpful to address these questions. The next 10 years will be necessary for scientific and technological advancement, but economic and political discussions need to move at the same speed, only then will the next 10 years be fruitful for patients.

## Figures and Tables

**Figure 1 jpm-11-01348-f001:**
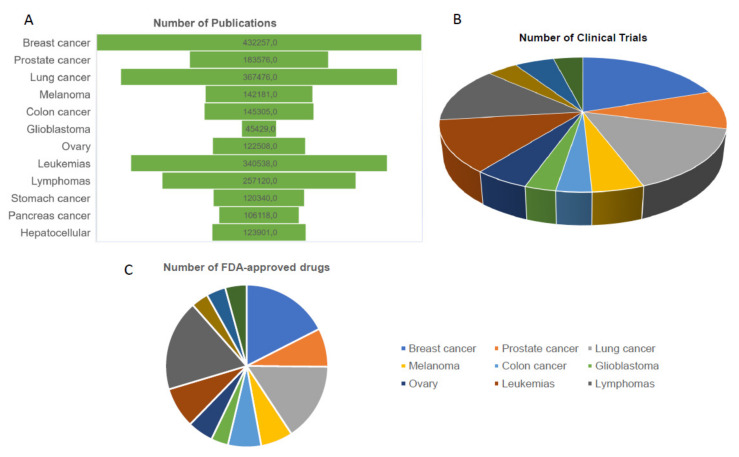
Number of publications (**A**), clinical trials (**B**) and FDA-approved drugs (**C**) in different types of cancers.

**Table 1 jpm-11-01348-t001:** Repositories of -Omics Data.

Database	Accesion Link	Main Features
The Cancer Genome Atlas (TCGA)	https://portal.gdc.cancer.gov/ (accessed on 11 October 2021)	Largest database of molecularly characterized over 20,000 primary cancer and matched normal samples spanning 33 cancer types.
ICGC Data Portal	https://dcc.icgc.org/ (accessed on 11 October 2021)	The ICGC Data Portal provides many tools for visualizing, querying, and downloading cancer data.
cBioPortal	www.cbioportal.org (accessed on 11 October 2021)	Graphical summaries; gene alteration; processed data; visualization.
UALCAN	http://ualcan.path.uab.edu/ (accessed on 11 October 2021)	Graphical summaries; processed data; visualization.
The human cancer metastasis database	hcmdb.i-sanger.com/index (accessed on 11 October 2021)	Integrated database designed to store and analyze large scale expression data of cancer metastasis.
Cancer Cell Line Encyclopedia	https://portals.broadinstitute.org/ccle (accessed on 11 October 2021)	Project for large-scale genetic characterization of ~1000 cancer cell lines.
MET500	https://met500.path.med.umich.edu/ (accessed on 11 October 2021)	Website for the MET500 metastatic cancer cohort: View aberrations in and interactions among gene sets and view the aberrations in a tumor sample, prioritized by their annotations.
COSMIC	http://cancer.sanger.ac.uk (accessed on 11 October 2021)	COSMIC, the Catalogue Of Somatic Mutations In Cancer, is the world’s largest and most comprehensive resource for exploring the impact of somatic mutations in human cancer.
MethyCancer	http://methycancer.psych.ac.cn (accessed on 11 October 2021)	Relationship among DNA methylation, gene expression and cancer.
SomamiR	http://compbio.uthsc.edu/SomamiR/ (accessed on 11 October 2021)	Correlation between somatic mutation and microRNA; genome-wide displaying.
UCSC Cancer Genomics Browser	https://genome-cancer.soe.ucsc.edu/ (accessed on 11 October 2021)	Clinical information; gene expression; copy number variation; visualization.
GDSC	http://www.cancerrxgene.org (accessedon 11 October 2021)	Drug sensitivity information; drug response information.
cansar	https://cansar.icr.ac.uk/ (accessed on 11 October 2021)	Multidisciplinary information; drug discovery.
NONCODE	http://www.noncode.org/ (accessed on 11 October 2021)	Data about ncRNAs; lncRNAs; up-to-date and comprehensive resource.
GEO DatSets	https://www.ncbi.nlm.nih.gov/gds (accessed on 11 October 2021)	This database stores curated gene expression DataSets, as well as original Series and Platform records in the Gene Expression Omnibus (GEO) repository.
Proteomics DB	https://www.proteomicsdb.org/ (accessed on 11 October 2021)	Repository of proteomics data, including 87 projects and 826 experiments.
The Humans Protein Atlas	https://www.proteinatlas.org/ (accessed on 11 October 2021)	The Human Protein Atlas was initiated in 2003 with the aim to map all the human proteins in cells, tissues and organs using an integration of various omics technologies, including antibody-based imaging, mass spectrometry-based proteomics, transcriptomics and systems biology.

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
