# Peer review of "Precision Medicine: Technological Impact into Breast Cancer Diagnosis, Treatment and Decision Making"

_jpm, 2021, doi:10.3390/jpm11121348_

Round 1

Reviewer 1 Report

Manuscript with the title ‘Precision Medicine: Technological impact into breast cancer diagnosis, treatment and decision making’ is a review paper whose aim is to present the most recent updates in the field of breast cancer. Author begins with a historic overview starting from Human genome project and sequencing of breast cancer tumour. Then she continuous with presenting cancer databases and targetable genomic alterations. The most detailed section deals with available tests, technical issues and feasibility. The last section is about precision oncology (clinical trials, drug discovery).

This review is well written and readable paper, except at the very end. Author failed to give some future predictions and estimations. Conclusion is somehow scarce and unfinished, so it should be corrected.

There are also few typing mistakes throughout the text (i.e. line 52, 92, 42…) that should be changed.

In my opinion, after minor corrections are fulfilled, manuscript will be suitable for publication.

Author Response

Dear Reviewer, 

Many thanks for your comments and suggestions. 

I agree with you about the conclusions. Considering your suggestion, I added a topic only regarding ‘Conclusions’ at page 25. In my opinion your suggestion improved the manuscript, I appreciated that.

I addressed all typing mistakes throughout the text as you pointed out.

I highlighted all the changes made to the manuscript to make it easier to follow the review.

Many Thanks, Tatiana

Reviewer 2 Report

Summary:

Breast cancer is the most common cancer in women and is second only to lung cancer. It has a long history of biomarkers that show both prognostic impact as well as define treatment paths with impact on clinical outcomes. For example, both hormone receptor status and HER2 amplification have moved from prognostic markers to well-established therapeutic targets. Tamoxifen was approved by the FDA in 1978 to treat metastatic, estrogen-positive breast cancer. Herceptin was first approved by the FDA in 1998 to treat HER2- positive metastatic disease. This success in translation and the incidence of the disease makes breast cancer an ideal tumor type to study in depth with each successive round of technological development. This review gives a brief history of how technological development has permitted more refined subtyping of breast tumors, delineates how that subtyping is now being used in various gene panels to refine prognosis and therapy choice, and speculates on the areas of research that are most likely to contribute to the advance of personalized treatment of breast cancer of the next 10 years. 

Broad Comments:

Strengths: This review covers a lot of ground, striving to put current clinical practice into the context of the broad body of research that informs it. It makes an excellent point that breast cancer shows how the efforts of many researchers and synthesis of that data can make a clinical impact. The manuscript highlights some of the recent innovations in clinical trials that are allowing bench science to be translated rapidly to the clinic. The review is most solid in defining what tests are currently available for clinicians to use to guide therapy, and although it doesn’t say it explicitly, also makes the point that current tests are focused on stratifying patients to prevent over-treatment. The manuscript does a reasonable job of defining both how the field has evolved with the technology and how that evolution led to the clinical tests available today. It also lays out, to some degree, where progress is likely to come in the next 10 years.

Weaknesses:

The logical organization of the history of technology advancement, application to breast cancer, and resulting improvements in therapeutic outcome is weak. This manuscript attempts to set out the history of precision medicine in breast cancer but fails to present the wealth of data that existed before The Cancer Genome Atlas project and the International Cancer Genomics Consortium began their data collection efforts until halfway through the manuscript. Furthermore, while I applaud the desire to showcase the many places where data currently exists, the section focusing on databases and repositories is out of place with the rest of the manuscript. The manuscript also fails to synthesize the results of the multitude of studies it cites, and it does not give citations for some studies it references. This prevents the author from identifying the key issues facing the ongoing implementation of precision medicine in breast cancer and more precisely defining the areas of opportunities for development over the next 10 years.

The logic of the review would be improved by rearranging the information in the first 3 sections and putting it in better historical context. Start with what prompted researchers to look for subtypes, and then describe how subtype identification was driven by advancement in the technology. My recommendation is to begin with the cytogenetic characterization of breast cancer in the late 1980’s and 1990s. The chapter entitled “Tumors of the Breast” by Teixiera, et al (Cancer Cytogenetics, ed. Heim and Mitelman, Wiley-Blackwell, 2015) is a good place to find these papers as Google or PubMed searching doesn’t always bring them up.  One of the first indications that breast cancer could be divided into genetically distinct subtypes came from a study published by Pandis, et al. in 1995 (Genes, Chromosomes, and Cancer 12(3):173-185). It established eight groups based on recurrent karyotype patterns. This didn’t make it into clinical practice because breast cancers are notoriously hard to karyotype. However, this study laid the foundation for the field to believe that gene expression studies would be fruitful in classifying breast cancer in meaningful ways. From this jumping off point, summarize the use of gene expression platforms to define clinically relevant subgroups and the key findings that pushed TCGA and the ICGC to begin the multi-omic profiling of tumors, breast cancer being prominently represented.

For the TCGA and ICGC studies, please make sure you have cited all the publications from where the results discussed come. In addition, synthesize the data, rather than reiterate detailed lists of genes and their mutational prevalence from a handful of studies. Which genes, across all studies, are recurrently mutated? At what rates and are there particular hotspots or mutational patterns that correlate with either subtype or point to potential therapeutic options, like the use of PARP inhibitors in tumors lacking robust homologous recombination? The manuscript can then move to the rare alterations and the discussion about how treatment is moving from solely tissue of origin as the primary criteria to molecular alteration. This section is very important and deserves a bit more attention and depth. The key strengths were highlighted, but what are the roadblocks to having all cancer care settings use multidisciplinary tumor boards to guide patient care? What needs to be in place over the next 10 years for this to become a reality?

The discussion of patents as it is currently could be removed completely without impacting the manuscript. Rather than give us the patents, synthesize in what areas recent patents suggest there will be commercial development and how that might impact clinical care. I think this was the purpose of this section, but it failed to do this well. Make a statement like “Improved drug delivery combined with new therapeutic modalities like siRNA represent one frontier for clinical advancement in breast cancer” and then support that contention with your discussion.

Most of the discussion in the what’s ahead section focuses not on precision medicine but on new treatments. The manuscript would be greatly improved if it discussed more in depth how the recent advances in immunotherapy are likely to be applied to breast cancer given that it is a cold tumor and what needs to happen in terms of precision medicine to make that work. In addition, what new technologies or current technologies applied at scale need to be implemented in order to make an impact on metastatic disease, and how the emerging interest in intra-tumor heterogeneity might play out in breast cancer should be addressed.

Specific Comments:

Page 9 line 351: The results of the TAILORx study were published in the New England Journal of Medicine in 2018 (DOI: 10.1056/NEJMoa1804710). Please update this discussion appropriately based on those results.

Page 9 line 372: The results of the MINDACT study were published in April 2021 in the Lancet 22(4):476-488. Please update this discussion appropriately based on those results.

Author Response

Dear Reviewer,

Many thanks for your suggestions and comments. I really appreciated that, in my opinion, the new version of the manuscript is improved.

I highlighted all the changes made to the manuscript to make it easier to follow the review.

Point 1.

Concerning your comment:  

‘The logic of the review would be improved by rearranging the information in the first 3 sections and putting it in better historical context. Start with what prompted researchers to look for subtypes, and then describe how subtype identification was driven by advancement in the technology. My recommendation is to begin with the cytogenetic characterization of breast cancer in the late 1980’s and 1990s. The chapter entitled “Tumors of the Breast” by Teixiera, et al (Cancer Cytogenetics, ed. Heim and Mitelman, Wiley-Blackwell, 2015) is a good place to find these papers as Google or PubMed searching doesn’t always bring them up.  One of the first indications that breast cancer could be divided into genetically distinct subtypes came from a study published by Pandis, et al. in 1995 (Genes, Chromosomes, and Cancer 12(3):173-185). It established eight groups based on recurrent karyotype patterns. This didn’t make it into clinical practice because breast cancers are notoriously hard to karyotype. However, this study laid the foundation for the field to believe that gene expression studies would be fruitful in classifying breast cancer in meaningful ways. From this jumping off point, summarize the use of gene expression platforms to define clinically relevant subgroups and the key findings that pushed TCGA and the ICGC to begin the multi-omic profiling of tumors, breast cancer being prominently represented.’

I reorganize the manuscript structure including your suggestions at pages 2 and 3. For sure, we can not forget the past and this historical view is very important.

Point 2.

Your comment: ‘For the TCGA and ICGC studies, please make sure you have cited all the publications from where the results discussed come. In addition, synthesize the data, rather than reiterate detailed lists of genes and their mutational prevalence from a handful of studies. Which genes, across all studies, are recurrently mutated? At what rates and are there particular hotspots or mutational patterns that correlate with either subtype or point to potential therapeutic options, like the use of PARP inhibitors in tumors lacking robust homologous recombination? The manuscript can then move to the rare alterations and the discussion about how treatment is moving from solely tissue of origin as the primary criteria to molecular alteration. This section is very important and deserves a bit more attention and depth. The key strengths were highlighted, but what are the roadblocks to having all cancer care settings use multidisciplinary tumor boards to guide patient care? What needs to be in place over the next 10 years for this to become a reality?’

I agree and this comment is valuable and enhances the content and it was helpful the to final conclusions.

In order to attend to your suggestions, I made alterations at pages 6,12 and 25.

Point 3.

Your comment: ‘The discussion of patents as it is currently could be removed completely without impacting the manuscript. Rather than give us the patents, synthesize in what areas recent patents suggest there will be commercial development and how that might impact clinical care. I think this was the purpose of this section, but it failed to do this well. Make a statement like “Improved drug delivery combined with new therapeutic modalities like siRNA represent one frontier for clinical advancement in breast cancer” and then support that contention with your discussion.’

Many thanks for your comprehension, the purpose of this section was to highlight commercial development. In the new version, this section was removed and the text was improved. You can see at page 24 and 25.

Point 4.

Your comment: ‘Most of the discussion in the what’s ahead section focuses not on precision medicine but on new treatments. The manuscript would be greatly improved if it discussed more in depth how the recent advances in immunotherapy are likely to be applied to breast cancer given that it is a cold tumor and what needs to happen in terms of precision medicine to make that work. In addition, what new technologies or current technologies applied at scale need to be implemented in order to make an impact on metastatic disease, and how the emerging interest in intra-tumor heterogeneity might play out in breast cancer should be addressed.’

In the new version, this section was improved and all your points were addressed. You can see at page 24 and 25.

Point 5.

Your comment: ‘Specific Comments:

Page 9 line 351: The results of the TAILORx study were published in the New England Journal of Medicine in 2018 (DOI: 10.1056/NEJMoa1804710). Please update this discussion appropriately based on those results.

Page 9 line 372: The results of the MINDACT study were published in April 2021 in the Lancet 22(4):476-488. Please update this discussion appropriately based on those results.’

All points addressed at pages 14 and 15.

Many thanks, Tatiana Tilli